# The Incidence and Risk Factors for Allogeneic Blood Transfusions in Pediatric Spine Surgery: National Data

**DOI:** 10.3390/healthcare11040533

**Published:** 2023-02-10

**Authors:** Justyna Fercho, Michał Krakowiak, Rami Yuser, Tomasz Szmuda, Piotr Zieliński, Dariusz Szarek, Grzegorz Miękisiak

**Affiliations:** 1Neurosurgery Department, Medical University of Gdansk, 80-214 Gdansk, Poland; 2Scientific Circle of Neurology and Neurosurgery, Neurosurgery Department, Medical University of Gdansk, 80-214 Gdansk, Poland; 3Department of Neurosurgery, Marciniaks Hospital, 54-094 Wroclaw, Poland; 4Institute of Medicine, Opole University, 45-052 Opole, Poland

**Keywords:** spine, surgery, scoliosis, deformity, blood transfusion, logistic regression

## Abstract

(1) Background: Pediatric spinal surgery is a blood-intensive procedure. In order to introduce a rational blood management program, identifying the risk factors for transfusions is mandatory. (2) Methods: Data from the national database covering the period from January 2015 to July 2017 were analyzed. The available data included the demographics, characteristics of the surgeries performed, length of stay, and in-house mortality. (3) Results: The total number of patients used for the analysis was 2302. The primary diagnosis was a spinal deformity (88.75%). Most fusions were long, with four levels or more (89.57%). A total of 938 patients received a transfusion; thus, the transfusion rate was 40.75%. The present study identified several risk factors; the most significant was a number of levels fused greater than 4 (RR 5.51; CI95% 3.72–8.15; *p* < 0.0001), followed by the deformity as the main diagnosis (RR 2.69; CI95% 1.98–3.65; *p* < 0.0001). These were the two most significant factors increasing the odds of a transfusion. Other factors associated with an increased risk of transfusion were elective surgery, the female sex, and an anterior approach. The mean length of stay in days was 11.42 (SD 9.93); this was greater in the transfused group (14.20 vs. 9.50; *p* < 0.0001). (4) Conclusions: The rate of transfusions in pediatric spinal surgery remains high. A new patient blood management program is necessary to improve this situation.

## 1. Introduction

Allogeneic transfusions are common in spinal surgery. The incidence of a perioperative red blood cell (RBC) transfusion varies between 8% and 36% [1]. In the pediatric population, this rate can be as high as 67% [2]. 

Although the absolute number of pediatric patients requiring large-volume transfusions is lower than adults [3], the specificity of spine surgery within this age group makes it particularly blood-intensive compared with other surgical fields [4]. A spinal deformity of any kind is the main indication for surgery in this age group. Such procedures are associated with a substantial blood loss because of the extensive soft-tissue dissection and instrumentation of the vertebrae [5]. Other factors (e.g., prolonged surgery, osteotomy requirements, and neuromuscular etiology) substantially increase the need for blood products. In a series of 107 patients who underwent posterior fusion surgery [6], neuromuscular etiology, a greater number of levels fused, and a lower body weight contributed to 53% of the variability in allogeneic transfusions, as shown by a multiple regression analysis.

Despite the best efforts to reduce the need for blood products, transfusions performed in infants and children are still associated with substantial risks. Compared with the adult population, there are considerable differences within the pediatric population, particularly the increased incidence of allergic reactions [7]. Other less common, severe transfusion reactions in this age group are hemolytic transfusion reactions [8], a transfusion-associated circulatory overload (TACO) [9], or a transfusion-related acute lung injury (TRALI) [10]; these have a prevalence of up to 1% [11]. There is evidence that children are more likely to be at a higher risk for non-infectious adverse events associated with transfusions than adults [12,13]. It has also been demonstrated that a liberal transfusion policy is associated with increased morbidity, even after controlling for possible confounders [1].

To make matters worse, the recent COVID-19 outbreak with the following lockdown further increased the pressure to review existing blood management strategies. It put a tremendous strain on the blood supply, and caused the worst blood shortage in over a decade worldwide [14]. Most significantly, the blend of multiple factors associated with governmental interventions preventing the virus has led to a significant drop in blood bank-based collections [15]. 

As it has become clear that transfusing RBCs may unnecessarily expose children to an increased risk without benefits, numerous multidisciplinary teams have begun to work on blood management programs specifically for children. One of the fundamental issues is the identification of procedures with an increased demand for blood products. Hence, the purpose of this study was to investigate the rate and risk factors of RBC transfusions in pediatric patients undergoing spinal surgery using a large national database.

## 2. Materials and Methods

### 2.1. Study Design

This was a retrospective cohort study.

### 2.2. Data

In this study, we present the results of an analysis of the database of the National Health Fund of Poland (Narodowy Fundusz Zdrowia, or NFZ). Since 2009, the NFZ has used diagnosis-related groups (DRGs) for reimbursement. It regularly publishes comprehensive data on hospitalizations on an official website [16].

The data used in this study covered all instrumented fusions funded by the NFZ from 1 January 2015 to 31 July 2017. The following DRGs were analyzed in patients < 18 years old: H51 (complex corrective spinal surgeries); and H52 (spinal procedures with implants). The above groups covered all relevant hospitalizations. As all spinal fusion procedures are fully reimbursed in Poland by the NFZ and because state-funded healthcare is generally reasonably accessible, it was safe to assume that the data covered nearly all cases of instrumented fusions in the pediatric population of the country. The ICD-9-CM code of 99.04 (Transfusion of Packed Cells) was used to identify the patients who received packed red blood cells. The following parameters were extracted from the database for a further analysis: gender, age, diagnosis, procedure performed (as per the ICD-9-CM code), approach (anterior/posterior), length of fusion (dichotomized into <4 levels/4 levels and up), length of stay (LOS), and in-house mortality.

### 2.3. Statistics

The continuous variables were compared using the *t*-test and the frequencies were compared using the χ^2^ test. The logistic regression model was used to identify the association between the blood transfusion and the selected factors. A statistical significance was defined as *p* < 0.05. Statistical analyses were carried out using StatsDirect, version 3.3.5 (StatsDirect Ltd., Merseyside, U.K.; https://www.statsdirect.com) and MedCalc, version 12.5.0.0 (MedCalc Software, Ostend, Belgium).

## 3. Results

Between 1 January 2015 and 31 July 2017, 2302 pediatric patients were discharged after surgery on the spine as the main indication for hospitalization. A total of 938 received a RBC transfusion of at least one unit.

### 3.1. Demographics and Operative Characteristics

The total number of surgeries per month had a notable seasonal variability, with nadirs at the turn of each year and peaks mid-year (Figure 1). The mean age was 13.4 years, with a significant female overrepresentation (68.59%; *p* < 0.0001). The complete demographic data are listed in Table 1. 

The mean volume of received packed RBCs was 650 mL (SD 230; range 200–4500). The principal diagnosis by far was a spinal deformity (88.75%), followed by trauma (7.95%), and then a tumor (1.17%). The vast majority of surgeries were elective (81.75%). Most fusions were long, with four levels or more (89.57%). An anterior approach was used in 4.04%. The in-house mortality was very low; only three deaths were noted (0.13%).

When comparing the transfused (*n* = 938) and non-transfused (*n* = 1364) groups, the former were younger (*p* < 0.001), with an even more significant proportion of females (*p* < 0.0013). In the transfused group, deformity patients comprised most of the patients requiring a transfusion (93.92% vs. 85.19%), with relatively less trauma. Elective surgeries were significantly overrepresented in the transfused group. Lastly, the proportion of more extensive (four levels and up) fusions was greater in the group that received allogeneic RBCs.

### 3.2. Postoperative Parameters

The mean LOS in days was 11.42 (SD 9.93) and was greater in the transfused group (14.20 vs. 9.50; *p* < 0.0001). This parameter was also greater in the group of patients who underwent long-fusion procedures vs. short-fusion (11.61 vs. 9.75; *p* < 0.001). The differences in mortality were insignificant; however, the overall low mortality precluded a meaningful analysis.

### 3.3. Risk Factors

The overall rate of transfusion was 40.75%. The most significant risk factor for a transfusion was 4 + levels (RR 5.51; CI95% 3.72–8.15; *p* < 0.0001), followed by the deformity as a diagnosis (RR 2.69; CI95% 1.98–3.65; *p* < 0.0001). Other factors associated with an increased risk of transfusion were elective surgery, the female sex, and an anterior approach. Trauma as the primary diagnosis was a significantly negative risk factor for a transfusion (RR 0.37; CI95% 0.26–0.53; *p* < 0.0001). The complete analysis is shown in Figure 2.

## 4. Discussion

Spinal surgery in the pediatric population is considered to be a blood-intensive procedure with an inherently high risk of a transfusion. The present study was designed to assess the transfusion incidence and risk factors among children who nationally underwent spinal surgery over a period of 31 months. It was propelled by the critical situation of blood banking in Poland caused by the COVID-19 pandemic. The overall transfusion rate in our series was 40.75%, which was significantly higher than the data from the literature. An article by Dick et al. [17] compared the rate of transfusion in adolescent idiopathic scoliosis patients from 2001 to 2015. They found that the proportion of patients transfused was 89.2% in 2001–2003 and 20.1% in 2013–2015. In another study from 2014 [18], which analyzed data from 43,983 pediatric patients operated on for idiopathic scoliosis, the overall allogeneic transfusion rate was 17.80%. A significant contributing factor to the high rate observed in Poland is a lack of a unified transfusion policy.

In recent years, the term ‘patient blood management’ (PBM) has gained popularity. It encompasses multidisciplinary interventions that focus on proactively identifying predisposing factors, reducing the requirements for a transfusion as well as reducing blood loss and improving hemostasis to maximize patient outcomes [19,20]. This initiative is quickly gaining momentum, and as many as 100 measures make up this approach [21].

In 2022, an expert group representing PBM organizations from the International Foundation for Patient Blood Management (IFPBM) proposed a global definition: “Patient blood management is a patient-centered and organized approach in which the entire health care team coordinates efforts to improve results by managing and preserving a patient’s own blood.” [22]. Most significantly, it reaches much further than a mere transfusion policy; although allogeneic transfusions can be an effective therapy for emergent life-threatening blood loss, it is important to assess the risks of blood loss and exert substantial preventive efforts to reduce this threat [23].

There are four main principles of PBM [24,25,26]:Anemia management: the detection and evaluation of the etiology, the stimulation of hematopoiesis, a reduction in oxygen consumption, and, finally, a transfusion when required;Blood conservation: a reduction in surgical invasiveness when feasible, the meticulous evaluation of blood loss, the use of all surgical techniques at hand to reduce blood loss, and the use of non-allogeneic transfusions (autologous transfusions, red blood cell salvage, and hemodilution);Optimization of hemostasis: the evaluation of hemostasis, the management of potential coagulopathy with targeted therapies, and the transfusion of blood when clinically indicated;Patient-centered care: the integration of patients into the PBM plan, informing patients, and working out possible solutions to any foreseeable circumstances.

The current PBM strategies are adopted mainly from the adult population as the pediatric literature is still limited [27]. The interventions can be divided into three groups based on the stage of treatment [28]. Preoperative strategies focus on augmenting the hemoglobin levels such as a reduction in blood draws, iron replacement, and the stimulation of erythropoiesis. Intraoperative PBM strategies include normovolemic hemodilution [29], red blood cell salvage, and antifibrinolytic agents [30]. The critical element of postoperative management is the application of adequate transfusion thresholds. The current expert consensus guidelines recommend a reasonable approach that considers the current clinical status. The threshold for pediatric patients (not including neonates) is 7 g/dl [31]. Significantly, in these patients (especially those weighing less than 20 kg), the allowable blood loss and transfusion volume should be calculated based on the weight and the target change in hemoglobin level [32].

Spinal surgery can significantly benefit from a rational PBM. A drastic reduction or even the mitigation of blood use can be achieved with an adequate policy [33]. One of the pillars of PBM is identifying the risk factors for receiving blood products. This study was designed to identify them within a large population of patients. The most significant risk factor for a transfusion as identified in the present study was a number of levels fused of four or more (RR 5.51). Other publications have reported similar findings with even higher cut-off values of 4 [34] or 10 [35]. Another risk factor revealed in this work was the diagnosis of scoliosis; a predominant indication for surgery in this age group. This finding was corroborated by Lam et al. in a national-level study from 2015 [36]. The same authors identified a trauma diagnosis as a protective factor, with an OR of 0.51 for a thoracolumbar injury and 0.33 for a cervical injury; this was similar to the present study. The female sex, another risk factor, was also observed by other authors [36,37]. Other major factors not investigated in this study were the neuromuscular type of deformity [38], a low BMI [39], and a higher Cobb angle [40]. In our study, the length of stay was longer for the transfused patients (14.20 vs. 9.50; *p* < 0.0001). However, this parameter was also greater in the group of patients who underwent more extensive procedures; thus, with the present data, it was impossible to determine if transfusions contributed to prolonged stays.

## 5. Study Limitations

The present study had limitations. As the main goal was to capture all cases treated nationwide, the precision of the study was traded with volume. The only reliable outcome measure was mortality and, to a certain extent, the LOS. No information was available on the hemoglobin/hematocrit levels at any point of the treatment. There was no information available on transfusion-related complications. Finally, the data were pooled and only predetermined statistics were available.

## 6. Conclusions

The incidence of an allogeneic blood transfusion in pediatric spine surgery remains high. The two most significant factors associated with an increased risk of transfusion were the length of fusion and the principal diagnosis of a spinal deformity. An introduction of rational PBM may decrease the demand for blood products.

## Figures and Tables

**Figure 1 healthcare-11-00533-f001:**
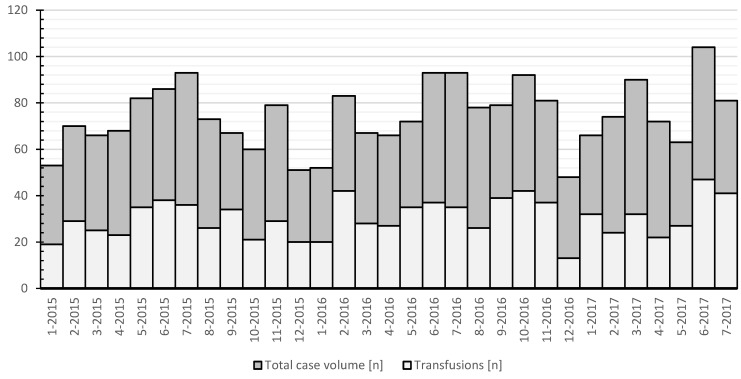
Case volume per each month. *X*-axis: number of patients (*n*); *Y*-axis: timeline (month/year).

**Figure 2 healthcare-11-00533-f002:**
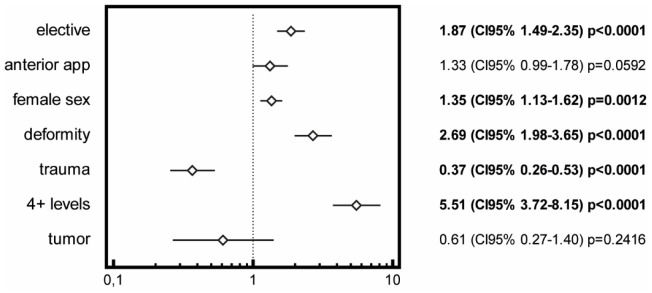
Forest plot displaying odds ratio (OR), 95% confidence interval, and *p*-values from regression models for the association between transfusions and selected factors.

**Table 1 healthcare-11-00533-t001:** Characteristics of patients in non-transfused vs. transfused groups.

Variable	All Patients (*n* = 2302)	Non-Transfused Group (*n* = 1364)	Transfused Group (*n* = 938)	*p*-Value
Demographics				
Age, years; mean	13.4	13.76	13.16	*p* < 0.001
Sex; number (%)				
Female	1579 (68.59)	900 (65.98)	679 (72.39)	*p* = 0.0013
Male	723 (31.41)	464 (34.02)	259 (27.61)	
Operative characteristics				
Deformity (%)	2043 (88.75)	1162 (85.19)	881 (93.92)	*p* < 0.0001
Trauma (%)	183 (7.95)	144 (10.56)	39 (4.16)	*p* < 0.0001
Tumor (%)	27 (1.17)	19 (1.39)	8 (0.85)	*p* = 0.3243
Elective surgery (%)	1882 (81.75)	1066 (78.15)	816 (86.99)	*p* < 0.0001
Anterior approach (%)	93 (4.04)	51 (3.74)	42 (4.48)	*p* = 0.1383
<4 Levels of fusion (%)	240 (10.43)	210 (15.4)	30 (3.2)	*p* < 0.0001
≥4 Levels of fusion (%)	2062 (89.57)	1154 (84.6)	908 (96.8)	
Postoperative parameters				
Length of stay	11.42	9.50	14.20	*p* < 0.0001
In-house mortality (%)	3 (0.13)	1 (0.07)	2 (0.21)	*p* = 0.7441
Repeated CRP (%)	150 (6.52)	71 (5.21)	79 (8.42)	*p* = 0.0028

## Data Availability

The data presented in this study are available on request from the corresponding author.

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
