# Peer review of "The Incidence and Risk Factors for Allogeneic Blood Transfusions in Pediatric Spine Surgery: National Data"

_healthcare, 2023, doi:10.3390/healthcare11040533_

Round 1

Reviewer 1 Report (Previous Reviewer 1)

I think the discussion should be more extensive and contain the methods for reducing the red blood cells transfusion rate.

The national database is extensive but has very few parameters to analyze.

Author Response

Dear Reviewer,

Thank you for your thorough review. As you requested, the discussion has been extended significantly. Of course, we agree that the data from our database was rather crude and stated this in the limitations' section

Kind regards,

Greg Miekisak

Reviewer 2 Report (New Reviewer)

Line 39, "compared to other surgical fields." the author must cite the references of these information. 

The study design is not clear in the materials and methods. Please write the type of the study research. Sub-sections may be suggested to follow up.

The legend of the Figure 1 needs to be on the same page as the figure. Legends for X- and Y-axis needs to be added.

Author Response

Dear reviewer,

below are detailed answers to your comments.

Kind Regards,

G. Miekisiak

Line 39, "compared to other surgical fields." the author must cite the references of these information. 

a reference has been added

The study design is not clear in the materials and methods. Please write the type of the study research. Sub-sections may be suggested to follow up.

study design was specified, sub-sections have been added

The legend of the Figure 1 needs to be on the same page as the figure. Legends for X- and Y-axis needs to be added.

Fixed

Reviewer 3 Report (New Reviewer)

Title

The incidence and risk factors for allogeneic blood transfusion in pediatric spine surgery: national data.

Critique

#1. The overall form of writing needs to be very modified. (Ex. Use duplicate Abstract words, highlight in yellow shades). Furthermore, English writing correction will be needed to understand readers.

#2. One of important factor in this study was length of stay. In national data, it is longer in transfusion group than non-transfusion group. This finding needs to be more illustrated in Discussion section. A more detailed description of whether this is the effect of high-level surgery (such as long-level surgery and/or osteotomy) or blood transfusion is needed.
#3. Blood transfusion is important, but how much blood volume was transfused into a patient is also important in the transfused group.

#4. Did the authors investigate the incidence of transfusion-related complications in the transfused group? I think such an investigation is essential for the "A new patient blood management program" mentioned in the conclusion.

#5. I believe that the research (allogenic blood transfusion in pediatric spine surgery) based on the national data is necessary and it has an archival value, which is a way forward to new blood management program that authors mentioned. However, the current results do not seem to suggest this direction.

Author Response

Dear reviewer,

Thank you for your thorough review. We did our best to address all issues. Below you will find a detailed response to each issue (blue).

#1. The overall form of writing needs to be very modified. (Ex. Use duplicate Abstract words, highlight in yellow shades). Furthermore, English writing correction will be needed to understand readers.

The manuscript had undergone substantial editing.

#2. One of important factor in this study was length of stay. In national data, it is longer in transfusion group than non-transfusion group. This finding needs to be more illustrated in Discussion section. A more detailed description of whether this is the effect of high-level surgery (such as long-level surgery and/or osteotomy) or blood transfusion is needed.

Relevant information was added to both results and discussion sections.

#3. Blood transfusion is important, but how much blood volume was transfused into a patient is also important in the transfused group.

We have added this information.

#4. Did the authors investigate the incidence of transfusion-related complications in the transfused group? I think such an investigation is essential for the "A new patient blood management program" mentioned in the conclusion.

Unfortunately, the data on transfusion-related complications was very scarce, and such an investigation – although an excellent idea – was not possible.

#5. I believe that the research (allogenic blood transfusion in pediatric spine surgery) based on the national data is necessary and it has an archival value, which is a way forward to new blood management program that authors mentioned. However, the current results do not seem to suggest this direction.

We have changed the manuscript and especially the discussion section. The main conclusion of our work is that the rate of transfusions in Poland is extremely high. We hope that diagnosing the problem is the crucial first step toward solving it.

Round 2

Reviewer 3 Report (New Reviewer)

It seems to have revealed the purpose of study more clearly. Thank you for taking the reviewer's opinions into consideration. In conclusion, this study have a sufficient academic value. 

This manuscript is a resubmission of an earlier submission. The following is a list of the peer review reports and author responses from that submission.

Round 1

Reviewer 1 Report

Line 82 - You mentioned about 679 (from 2302) patients who received at least one unit of RBC, but i line 92 transfused group contains 938 patient. How do you explain this difference?

lines 150-158 conclusion about the infection rate is going too far. CRP can be elevated due to surgical procedure, without any clinical signs of infection. Are there any data about the cases of infection (SSI or systemic) in the NFZ database?

It's good to mention about the weak points of the paper.

Author Response

Dear reviewer, 

Thank you for your meticulous revision. We’ve tried to address all issues in the text below. 

G Miekisiak 

 Line 82 - You mentioned about 679 (from 2302) patients who received at least one unit of RBC, but i line 92 transfused group contains 938 patient. How do you explain this difference? 

 @You are correct, it was 679 FEMALES and 938 TOTAL – it has been fixed. 

lines 150-158 conclusion about the infection rate is going too far. CRP can be elevated due to surgical procedure, without any clinical signs of infection. Are there any data about the cases of infection (SSI or systemic) in the NFZ database? 

 @Agreed, this conclusion is a bit far-fetched, we changed the paragraph accordingly. 

 It's good to mention about the weak points of the paper. 

@The paragraph has been added. 

Reviewer 2 Report

This is a retrospective study evaluating the incidence and risk factors for allogenic blood transfusion in pediatric spine surgery using a national database registry.  The authors used a national database to identify pediatric patients undergoing spine surgery. They identified 2302 patients and determined which patients received a transfusion using a ICD-9-CM code.  They then compared factors between the transfused and non-transfused group.  They concluded the overall rate of transfusion was 40.75% in this patient population.  They then identified risk factors for transfusion were >4 levels fused, diagnosis of deformity, elective surgery, female sex, and anterior approach.  The authors unfortunately did not perform a multivariable analysis to determine if any of these factors were independent risk factors for transfusion.

While this paper has potential to add to the knowledge base regarding risk factors for transfusions in the pediatric spine populations, there are major concerns with this paper that require a major revision prior to accepting for publication.

1) Word choice and syntax: There are awkward phrases that do not add meaning to the manuscript.  There are also awkward phrases that are not specific – “blood-intensive” “deformity”

2) Analysis: The authors essential performed a univariate analysis to determine risk factors for blood transfusion.  However, there is an opportunity to perform a multivariable analysis to determine which of these factors are independently associated with transfusions.

3) Study Design: It is not clear how the authors which factors should be included.  However, “deformity” is a vague term that as demonstrated by this study includes the majority of their patient population. The authors instead should try to determine if they can divide patients with a scoliosis diagnosis into “neuromuscular” and “idiopathic” as this tends to be where the differences in both blood transfusions lie.  I would then compare the differences in idiopathic and neuromuscular scoliosis transfusion rates.  Finally, it is not clear why the authors used CRP instead of something like Hematocrit or Hemoglobin as either an outcome or risk factor.

4) Discussion: The discussion is much more well written than the Introduction and addresses a lot of the issues with awkward syntax and word choice.  Please review the introduction.  The discussion does not address limitations of this study.  This would include information such as a national registry database without access to other details of the population, etc.

Introduction: Overall, there are several awkward and non-specific statements throughout the introduction that make it difficult to read. The authors site on the COVID-19 global pandemic as a reason to evaluate the blood usage in pediatric patients.  However, the study period is 2015-2017.

Page 1, line 24: despite efforts: it is not clear what efforts (if any are taken). Perhaps remove this statement or please clarify.

Page 1, line 29: What does blood intensive refer to? Please clarify.

Page 1, line 29: The reference to the deformity is an awkward statement and not clear exactly what this means as the “main indication” for surgery

Page 1, line 35: It is not clear what “53% of variability in allogeneic transfusion” means.  What is the variability and does this refer to whether patients are transfused or not or does it refer to the variability in rates of transfusion in this population

Page 1, line 36: Please clarify “best efforts” – does this refer to improving safety of blood transfusions or reducing blood transfusions? 

Page 1, line 36-42: The mentioned adverse events are actually pretty rare in the pediatric population.  Febrile allergic hypotensive reactions tend to be more common in pediatric patients and this should be referenced.

Page 1, line 43, Page 2-48. The authors have an entire paragraph referring to the global pandemic and blood shortages as the reason for this study, yet the study period is 2015-2017.  Would recommend focusing your introduction on risk of blood transfusions in pediatric spine surgery patients and factors that have contributed to transfusions.

Methods:

Page 2, line 66: Please remove the word virtually

Page 2, line 67, please indicate if these are packed “red” blood cells

Results:

These results are all independent risk factors.  In general, this seems like a univariate analysis for risk factors that predict transfusion.  However, these results should be put into a multivariable analysis to determine if any of these factors that are significant from the univariate analysis are predictive of transfusions.  For example, if there are older patients with less levels fused in the majority of the trauma patients, this might explain why trauma was a negative predictive factor for transfusion. 

Page 2, line 82. Please add a period

Page 2, line 87 – does deformity refer to scoliosis?

Page 2, line 91 ( please consider adding percentage of deaths)

Page 2, line 94 – please remove “a” in front of transfused group

Page 2, line 94, consider rephrasing to “patients with scoliosis comprised the majority of patients requiring a transfusion”. 

Page 3, line 96: please rephrase this sentence. The word longer is confusing – is it a longer section of the spine or longer operative times? Reconsider how to phrase this.

Page 3, line 99 – please indicate the units of LOS (I assume days but this is not specficied).

Page 3, line 100 – please also address in the methods. Why were you looking at CRP levels in the patient group? It seems that Hematocrit or Hemoglobin levels are more relevant to transfusions.

Author Response

Dear reviewer, 

Thank you for your meticulous revision. We’ve tried to address all issues in the text below. 

G Miekisiak 

This is a retrospective study evaluating the incidence and risk factors for allogenic blood transfusion in pediatric spine surgery using a national database registry.  The authors used a national database to identify pediatric patients undergoing spine surgery. They identified 2302 patients and determined which patients received a transfusion using a ICD-9-CM code.  They then compared factors between the transfused and non-transfused group.  They concluded the overall rate of transfusion was 40.75% in this patient population.  They then identified risk factors for transfusion were >4 levels fused, diagnosis of deformity, elective surgery, female sex, and anterior approach.  The authors unfortunately did not perform a multivariable analysis to determine if any of these factors were independent risk factors for transfusion. 

While this paper has potential to add to the knowledge base regarding risk factors for transfusions in the pediatric spine populations, there are major concerns with this paper that require a major revision prior to accepting for publication. 

1) Word choice and syntax: There are awkward phrases that do not add meaning to the manuscript.  There are also awkward phrases that are not specific – “blood-intensive” “deformity” 

2) Analysis: The authors essential performed a univariate analysis to determine risk factors for blood transfusion.  However, there is an opportunity to perform a multivariable analysis to determine which of these factors are independently associated with transfusions. 

3) Study Design: It is not clear how the authors which factors should be included.  However, “deformity” is a vague term that as demonstrated by this study includes the majority of their patient population. The authors instead should try to determine if they can divide patients with a scoliosis diagnosis into “neuromuscular” and “idiopathic” as this tends to be where the differences in both blood transfusions lie.  I would then compare the differences in idiopathic and neuromuscular scoliosis transfusion rates.  Finally, it is not clear why the authors used CRP instead of something like Hematocrit or Hemoglobin as either an outcome or risk factor. 

4) Discussion: The discussion is much more well written than the Introduction and addresses a lot of the issues with awkward syntax and word choice.  Please review the introduction.  The discussion does not address limitations of this study.  This would include information such as a national registry database without access to other details of the population, etc. 

Introduction: Overall, there are several awkward and non-specific statements throughout the introduction that make it difficult to read. The authors site on the COVID-19 global pandemic as a reason to evaluate the blood usage in pediatric patients.  However, the study period is 2015-2017. 

 Page 1, line 24: despite efforts: it is not clear what efforts (if any are taken). Perhaps remove this statement or please clarify. 

 @Removed 

 Page 1, line 29: What does blood intensive refer to? Please clarify. 

 @A clarification was added 

Page 1, line 29: The reference to the deformity is an awkward statement and not clear exactly what this means as the “main indication” for surgery 

 @Spinal deformity of any kind – the text was edited 

Page 1, line 35: It is not clear what “53% of variability in allogeneic transfusion” means.  What is the variability and does this refer to whether patients are transfused or not or does it refer to the variability in rates of transfusion in this population 

 @This values come from the multiple regression analysis – text has been changed 

Page 1, line 36: Please clarify “best efforts” – does this refer to improving safety of blood transfusions or reducing blood transfusions?  

 @best efforts to reduce the need for blood products – the sentence has been changed 

Page 1, line 36-42: The mentioned adverse events are actually pretty rare in the pediatric population. Febrile allergic hypotensive reactions tend to be more common in pediatric patients and this should be referenced. 

 @That is a valid point. We added a significant reference (in our view) 

Page 1, line 43, Page 2-48. The authors have an entire paragraph referring to the global pandemic and blood shortages as the reason for this study, yet the study period is 2015-2017.  Would recommend focusing your introduction on risk of blood transfusions in pediatric spine surgery patients and factors that have contributed to transfusions. 

 @The rationale for this study was to indicate the most sensitive, blood-thirsty procedures within orthopedics. The paragraph was changed; if it is still not to your liking, we will remove it completely. 

 Methods: 

Page 2, line 66: Please remove the word virtually 

 @Removed 

 Page 2, line 67, please indicate if these are packed “red” blood cells 

@added 

 Results: 

These results are all independent risk factors.  In general, this seems like a univariate analysis for risk factors that predict transfusion.  However, these results should be put into a multivariable analysis to determine if any of these factors that are significant from the univariate analysis are predictive of transfusions.  For example, if there are older patients with less levels fused in the majority of the trauma patients, this might explain why trauma was a negative predictive factor for transfusion.  

 @We used a univariate multivariable analysis for our model, and we considered the variables independent (although, in truth, they are slightly dependent). This is the guide I use extensively in this area:

https://academic.oup.com/ntr/article/23/8/1446/5812038 

Page 2, line 82. Please add a period 

@added 

Page 2, line 87 – does deformity refer to scoliosis? 

@Yes, the term was expanded 

Page 2, line 91 ( please consider adding percentage of deaths) 

@done 

Page 2, line 94 – please remove “a” in front of transfused group 

@done 

Page 2, line 94, consider rephrasing to “patients with scoliosis comprised the majority of patients requiring a transfusion”.  

@done 

Page 3, line 96: please rephrase this sentence. The word longer is confusing – is it a longer section of the spine or longer operative times? Reconsider how to phrase this. 

@changed 

Page 3, line 99 – please indicate the units of LOS (I assume days but this is not specified). 

@changed 

Page 3, line 100 – please also address in the methods. Why were you looking at CRP levels in the patient group? It seems that Hematocrit or Hemoglobin levels are more relevant to transfusions. 

 @The only available data was a simple binary fact that the assay was performed. Not the value. We realize that the conclusion based in the mere fact of repeated testing was far-reaching, and now we’ve made it clear in the text. We’ve had no information on levels of hematocrit/hemoglobin whatsoever. 

Reviewer 3 Report

In this manuscript, Miekisiak et al. performed a retrospective study assessing the incidence and risk factors for allogeneic blood transfusions in pediatric spinal surgery. They found that the overall transfusion rate in their country was 41%, and they also identified a few risk factors for transfusion in this population. Please find below comments to improve the quality of the manuscript.

Entire Manuscript: I question the novelty of the manuscript, as many papers have already published data on the incidence and risk factors of transfusion in pediatric spine surgery. Could the authors address this question of how this manuscript adds to the transfusion literature?

Entire Manuscript: The word “allogeneic” is mis-spelled throughout the entire manuscript (including in the title). Please amend this error. “Allogenic” blood is incorrect.

Abstract: This abstract needs to be significantly expanded upon to better reflect the manuscript. The methods require information on the study population, number of patients, primary outcomes, statistical analysis etc. The results require more statistical data, including odds ratios, P-values etc. The abstract needs to be significantly re-worked.

Methods P2 L55-60: Were there any exclusion criteria in this study? I encourage the authors to include a flow chart figure outlining the total eligible patient population, the inclusion and exclusion criteria, and the final patient population.

Methods P2 L65-70: Could the authors confirm that only intraoperative transfusions were included (not postoperative)?

Methods P2 L65-70: Since these are pediatric patients, were any of them transfused in a weight-based manner (~10 cc/kg)? This important point needs to be clarified and addressed in the manuscript. Related to this, please provide the range of ages of the patients in this study.

Results P2 L85-90: Was any stratification performed based on severity of surgery? Given that all spinal fusion procedures were included, the authors should stratify based on the number of fusion levels/severity of procedure.

Results P3 L103-109: Is there any data on the pre-transfusion Hb for these patients? That is one of the most important predictors of risk of transfusion. In addition, is there any data on the total volume of blood patients received?

Results P3 L109: Is there any data on transfusion-related complications?

Results P3 L109: An overwhelming majority of the patient population in the transfused group (~90%) were undergoing procedures for deformity with >4 levels of fusion. As a result, the authors’ conclusion that these factors were associated with an increased risk of transfusion is somewhat biased given that nearly all patients receiving a transfusion met these criteria.

Discussion P5 L24: Were any patient blood management techniques implemented during the time frame of this study? The authors acknowledge their transfusion rate is significantly higher than other studies – why is that? What are the transfusion policies in Poland?

Discussion P5 L150-160: I significantly question the author’s statement that repeated CRPs can be used in isolation as a surrogate for infection. This is a stretch without additional clinical and laboratory data, as CRPs can be elevated for a multitude of reasons. I would encourage deleting this paragraph from the Discussion, as the clinical rationale is limited.

Discussion P5 L158: Please include a paragraph outlining the limitations of this study.

Author Response

Dear reviewer, 

Thank you for your meticulous revision. We’ve tried to address all issues in the text below. 

G Miekisiak 

 In this manuscript, Miekisiak et al. performed a retrospective study assessing the incidence and risk factors for allogeneic blood transfusions in pediatric spinal surgery. They found that the overall transfusion rate in their country was 41%, and they also identified a few risk factors for transfusion in this population. Please find below comments to improve the quality of the manuscript. 

Entire Manuscript: I question the novelty of the manuscript, as many papers have already published data on the incidence and risk factors of transfusion in pediatric spine surgery. Could the authors address this question of how this manuscript adds to the transfusion literature? 

 @This is a population-based study covering ALL cases of spinal instrumented fusion in the pediatric population in a large European country with a population of close to 40 million. What makes it particularly interesting is that it  

 Entire Manuscript: The word “allogeneic” is mis-spelled throughout the entire manuscript (including in the title). Please amend this error. “Allogenic” blood is incorrect. 

@Fixed 

 Abstract: This abstract needs to be significantly expanded upon to better reflect the manuscript. The methods require information on the study population, number of patients, primary outcomes, statistical analysis etc. The results require more statistical data, including odds ratios, P-values etc. The abstract needs to be significantly re-worked. 

 @It has been rewritten substantially 

Methods P2 L55-60: Were there any exclusion criteria in this study? I encourage the authors to include a flow chart figure outlining the total eligible patient population, the inclusion and exclusion criteria, and the final patient population. 

@There were no exclusion criteria, we looked at the entire population of children who underwent instrumented spinal fusion. Thus, the flow chart would make no sense – with only two arms (transfused, non-transfused), and no exclusions. 

Methods P2 L65-70: Could the authors confirm that only intraoperative transfusions were included (not postoperative)? 

 @All transfusions within the treating unit were included, perioperative and postoperative. 

 Methods P2 L65-70: Since these are pediatric patients, were any of them transfused in a weight-based manner (~10 cc/kg)? This important point needs to be clarified and addressed in the manuscript. Related to this, please provide the range of ages of the patients in this study. 

 @There is on the transfusion manner in particular patients, however, it is common practice in Poland to transfuse 10-15-20 ml/kg in children depending on clinical situation/urgency 

 Results P2 L85-90: Was any stratification performed based on severity of surgery? Given that all spinal fusion procedures were included, the authors should stratify based on the number of fusion levels/severity of procedure. 

 @We only had data on the length of fusion and the type of approach – anterior and/or posterior. 

 Results P3 L103-109: Is there any data on the pre-transfusion Hb for these patients? That is one of the most important predictors of risk of transfusion. In addition, is there any data on the total volume of blood patients received? 

@There was no data on the Hgb level available, the volume received was expressed in the number of units. 

 Results P3 L109: Is there any data on transfusion-related complications? 

 @There was no data on transfusion-related complications. 

Results P3 L109: An overwhelming majority of the patient population in the transfused group (~90%) were undergoing procedures for deformity with >4 levels of fusion. As a result, the authors’ conclusion that these factors were associated with an increased risk of transfusion is somewhat biased given that nearly all patients receiving a transfusion met these criteria. 

@This is a valid point, however, the sheer volume of patients (2302) allowed for meaningful analysis   

Discussion P5 L24: Were any patient blood management techniques implemented during the time frame of this study? The authors acknowledge their transfusion rate is significantly higher than other studies – why is that? What are the transfusion policies in Poland? 

@There is no unified policy in Poland, the decisions are made in a case-by-case basis. That is one of the conclusions that the lack of policy leads to a very high rate of transfusions compared with other countries. 

Discussion P5 L150-160: I significantly question the author’s statement that repeated CRPs can be used in isolation as a surrogate for infection. This is a stretch without additional clinical and laboratory data, as CRPs can be elevated for a multitude of reasons. I would encourage deleting this paragraph from the Discussion, as the clinical rationale is limited.  

@This is a controversial statement, agreed. The paragraph has been rewritten but if you still think it is too far-reaching we will delete it.  

Discussion P5 L158: Please include a paragraph outlining the limitations of this study. 

@The paragraph has been added, consisting mostly of the points you have raised – thank you for that. 

Round 2

Reviewer 3 Report

I appreciate the author's responses, but unfortunately there remain significant concerns in study design and many of my concerns have not been addressed during this revision. 

It still remains unclear how this manuscript is novel, as the authors did not address my initial comment. The word "allogeneic" continues to be spelled incorrectly throughout the manuscript (and in the title). I am also surprised there were zero exclusion criteria in this study. How did the authors account for elective vs. emergent surgery - was a subgroup analysis performed? Was cell saver used in any of these cases? What about TXA? Also, did any patients receive FFP or PLT? All this information is required to better contextualize the results. Furthermore, is there any data on how much blood patients received -i.e. how many units? Unfortunately, in large studies like this, a lot of pertinent information (pre-transfusion Hb, reasons for transfusion, etc.) are not available, which confounds the results.

The authors state that it is common practice in Poland to transfuse 10-15-20 ml/kg in children. Were any of the patients in this study transfused in a weight based manner, and if so how were they incorporated? All the results in this study seem to be in RBC units, not volume, so how was this reconciled in the study?

The authors have also failed to address another concern, which was that an overwhelming majority of the patient population in the transfused group (~90%) were undergoing procedures for deformity with >4 levels of fusion. As a result, the conclusions are slightly biased given that nearly all patients met these criteria. This would bias results regardless of the size of the patient population. Can the authors stratify within the >4 levels of fusion, in other words compare results based on 4 levels, 5 levels, 6 levels etc.

The CRP paragraph is still too far-reaching. The authors even claim that "this conclusion can be considered far-fetched as there is no further evidence to support it." Please delete this paragraph, as one should not overstate their conclusions of a study.